# Controlling Unconventional Secretion for Production of Heterologous Proteins in *Ustilago maydis* through Transcriptional Regulation and Chemical Inhibition of the Kinase Don3

**DOI:** 10.3390/jof7030179

**Published:** 2021-03-03

**Authors:** Kai P. Hussnaetter, Magnus Philipp, Kira Müntjes, Michael Feldbrügge, Kerstin Schipper

**Affiliations:** Institute for Microbiology, Heinrich Heine University Düsseldorf, Universitätsstraße 1, 40225 Düsseldorf, Germany; kai.hussnaetter@uni-duesseldorf.de (K.P.H.); Magnus.Philipp@uni-duesseldorf.de (M.P.); kira.muentjes@hhu.de (K.M.); michael.feldbruegge@hhu.de (M.F.)

**Keywords:** autoinduction, chemical genetics, cytokinesis, inducible promoter, nanobody, regulated secretion, unconventional secretion, *Ustilago maydis*

## Abstract

Heterologous protein production is a highly demanded biotechnological process. Secretion of the product to the culture broth is advantageous because it drastically reduces downstream processing costs. We exploit unconventional secretion for heterologous protein expression in the fungal model microorganism *Ustilago maydis*. Proteins of interest are fused to carrier chitinase Cts1 for export via the fragmentation zone of dividing yeast cells in a lock-type mechanism. The kinase Don3 is essential for functional assembly of the fragmentation zone and hence, for release of Cts1-fusion proteins. Here, we are first to develop regulatory systems for unconventional protein secretion using Don3 as a gatekeeper to control when export occurs. This enables uncoupling the accumulation of biomass and protein synthesis of a product of choice from its export. Regulation was successfully established at two different levels using transcriptional and post-translational induction strategies. As a proof-of-principle, we applied autoinduction based on transcriptional *don3* regulation for the production and secretion of functional anti-Gfp nanobodies. The presented developments comprise tailored solutions for differentially prized products and thus constitute another important step towards a competitive protein production platform.

## 1. Introduction

Recombinant proteins are ubiquitous biological products with versatile industrial, academic, and medical applications [1,2]. Well-established hosts for protein production include, e.g., bacteria such as *Escherichia coli* [3], yeasts such as *Saccharomyces cerevisiae* or *Pichia pastoris* [2,4] or mammalian and insect tissue cultures [5,6]. Importantly, the nature of a protein largely influences the choice of a particular expression system, and not every protein is adequately expressed in the standard platform of choice [7]. Thus, there is not a universal protein expression system and the demand for alternative production hosts is increasing. In general, secretory systems are advantageous because the protein product is exported into the medium allowing for economic and straightforward downstream processing workflows [8]. Due to their extraordinary secretion capacities and inexpensive cultivation, fungal expression hosts are promising candidates for novel platforms and already the preferred hosts for the production of proteases and other hydrolytic enzymes [9,10]. However, the synthesis of heterologous proteins still imposes major challenges in fungal expression hosts [11]. One reason is the occurrence of atypical post-translational modifications during conventional secretion via the endomembrane system [12]. Furthermore, secreted fungal proteases are often destructive to the exported products [9,13]. Hence, it is important to further develop tailor-made strategies to provide a broad repertoire of potent fungal host organisms and enable the economic production of all relevant requested proteins in their functional form.

In the past years, we have established heterologous protein production based on unconventional chitinase secretion in the fungal model microorganism *Ustilago maydis* [14,15,16,17]. The phenomenon of unconventional secretion has been described for an increasing number of eukaryotic proteins [18,19]. Well-characterized examples include mammalian fibroblast growth factor 2 which is released via self-sustained translocation [20,21] and acyl-CoA binding protein Acb1 exported via specialized compartments of unconventional secretion (CUPS) [22]. However, in most other cases detailed mechanistic insights are still lacking. Furthermore, biotechnological applications for these systems have been proposed [23] but have not been described to date. 

In our system, chitinase Cts1 is used as a carrier for export of proteins of interest. The main advantage of this unique system is that proteins do not have to pass the endomembrane system as they would during conventional secretion. This circumvents post-translational modifications such as *N*-glycosylation and other drawbacks such as size limitations of the endomembrane system. Since non-natural *N*-glycosylation of proteins can be destructive to their activity [12,24] unconventional secretion is a good choice for sensitive proteins such as those originating from bacteria [25]. Bacterial β-glucuronidase (Gus) for example cannot be secreted in an active form via the conventional pathway [12]. By contrast, Cts1-mediated unconventional secretion results in active protein in the culture supernatant. As a versatile reporter, Gus is therefore also perfectly suited to detect and quantify unconventional secretion [14,26]. The applicability of the expression system has been shown by successful production of several functional proteins such as single-chain variable fragments (scFvs), nanobodies, or different bacterial enzymes such as Gus, β-galactosidase (LacZ), or polygalacturonases [14,25,27,28]. 

Recently, we obtained the first insights into the cellular mechanism of unconventional secretion [29,30,31]. During cytokinesis of yeast cells, a primary septum is formed at the mother cell side, followed by a secondary septum at the daughter cell side, delimiting a so-called fragmentation zone (Figure 1A) [32]. Upon formation of the daughter cell, Cts1 is targeted to this zone and likely functions in degradation of the remnant cell wall to separate mother and daughter (Figure 1B). Here, it acts in concert with a second, conventionally secreted chitinase, Cts2 [33]. Genetic screening identified the potential anchoring factor Jps1, a yet undescribed protein that exhibits an identical localization as Cts1 and is crucial for its export (Figure 1C) [30]. In addition, the presence of two proteins required for secondary septum formation, guanine nucleotide exchange factor (GEF) Don1 and germinal center kinase Don3 (Figure 1D), is essential for Cts1 secretion. Loss of either protein involved in septum formation results in the formation of cell aggregates and a strongly diminished extracellular chitinase activity [31]. This suggested a lock-type mechanism for Cts1 secretion [29]. Interestingly, Don3 itself was also found to be released similar to Cts1 [31].

Here, we established for the first-time regulatory mechanisms for protein production by unconventional secretion, which are based on our recent insights into the export pathway. Efficient regulation was achieved by two basic strategies: (i) transcriptional and (ii) post-translational induction of the previously identified unconventional secretion factor Don3. This led to new regulatory options including an autoinduction process, which can be applied depending on the need of the product of interest.

## 2. Materials and Methods

### 2.1. Molecular Biology Methods

All plasmids (pUMa vectors) generated in this study were obtained using standard molecular biology methods established for *U. maydis* including Golden Gate cloning [34,35,36]. Genomic DNA of *U. maydis* strain UM521 was used as template for PCR reactions. The genomic sequence for this strain is stored at the EnsemblFungi database [37]. All plasmids were verified by restriction analysis and sequencing. Oligonucleotides applied for cloning are listed in Table 1. The generation of pUMa3329_Δupp1_P_crg_-eGfp-Tnos-natR, pUMa2113_pRabX1-P_oma__gus-SHH-cts1, pUMa2240_Ip_P_oma_-his-anti-Gfpllama-ha-Cts1-CbxR and pUMa2775_um03776D_hyg had been previously described [14,28,30,31] but often used in differing strain backgrounds in the present study (for references see Table 2). For generation of pUMa4234_Δupp1_P_crg_-jps1-eGfp-Tnos-natR and pUMa4235_Δupp1_P_crg_-jps1-Tnos-natR, *jps1-gfp* or *jps1* were amplified and inserted into an *upp1* insertion vector. Therefore, pUMa3330 [31] was digested using MfeI and AscI, serving as cloning backbone. A PCR product obtained with primer combination oUM910/oUM912 for *jps1-gfp* or oUM910/oUM911 for *jps1* using pUMa3095 [30] as a template, was inserted into the digested backbone. For generation of pUMa4308_Δupp1_P_crg_-don3(M157A)-Tnos-natR and pUMa4313_Δupp1_P_crg_-don3(M157A)-eGfp-Tnos-natR site-directed mutagenesis using primer pair oAB23/oAB24 was performed on plasmids pUMa3331 or pUMa3330 [38], respectively, resulting in exchange of a single base pair [38]. For generation of pUMa3293_pPjps1—jps1-eGfp_CbxR, *jps1* promoter was amplified using primer combination oUP65/oUP66, *jps1* was amplified using primer combination oMB190/oMB520, *eGfp* was amplified using primer combination MB521/oMB522. PCR products were digested using BamHI, EcoRI, NotI, NdeI and inserted in the digested backbone pUMa2113 [27]. Detailed cloning strategies and vector maps will be provided upon request.

### 2.2. Strain Generation

*U. maydis* strains used in this study were obtained by homologous recombination yielding genetically stable strains (Table 2) [39]. All strains were derived from strain AB33. In this laboratory strain, the *b* mating type locus has been manipulated by insertion of compatible *b* genes controlled by a nitrogen-inducible promoter. This allows for a switch between yeast and filamentous growth by use of different nitrogen sources in the cultivation medium [40]. Genomic integrations were positioned either at the *ip* or the *upp1* locus, two established loci for genomic integrations. The *ip* locus encodes an iron-sulfur protein of the respiratory chain. Exchanging a single amino acid in this enzyme renders the cells resistant against the antibiotic carboxin [41]. Plasmids carrying the *ip^R^* gene mediating carboxin resistance were used and integrated in the native *ip^S^* locus of carboxin sensitive strains [14]. For transformation, these integrative plasmids were digested within the *ip^R^* region using the restriction endonuclease SspI, resulting in a linear DNA fragment. For insertions at the *upp1* locus (*umag_02178*) [27], plasmids harbored a nourseothricin resistance cassette and the integration sequence, flanked by homologous regions for the respective insertion locus. For transformation, the insertion cassette was excised from the plasmid backbone using SspI or SwaI [36]. For generation of deletion mutants, hygromycin resistance cassette containing constructs flanked by regions homologous to the 5′and 3’ sequences of the genes to be deleted were used. Again, deletion cassettes were excised from plasmid backbones prior to transformation [36]. For all genetic manipulations, *U. maydis* protoplasts were transformed with linear DNA fragments for homologous recombination. All strains were verified by Southern blot analysis [39]. The *upp1* locus encodes the secreted aspartatic protease Upp1. Along with other genes for secreted proteases *upp1* can be deleted without causing any morphologic phenotype while the proteolytic activity in the culture supernatant is reduced and heterologous proteins are stabilized [27]. For *upp1* insertion, digoxigenin-labelled probes were obtained by PCR using primer combinations oRL946/oRL947 and oRL948/oRL949 on template pUMa1538 [27]. For in locus modifications the flanking regions were amplified as probes. For *ip* insertions, the probe was obtained by PCR using the primer combination oMF502/oMF503 and the template pUMa260 [42]. Primer sequences are listed in Table 1.

### 2.3. Cultivation

*U. maydis* strains were cultivated at 28 °C in complete medium (CM) supplemented [44] with 1% (*w*/*v*) glucose (CM-glc) or with 1% (*w*/*v*) arabinose (CM-ara) if not described differently or in YepsLight [45]. CM cultures were eventually buffered with 0.1 M MES as mentioned in the respective section. Solid media were supplemented with 2% (*w*/*v*) agar. Growth phenotype and Gfp fluorescence in different media was evaluated using the BioLector microbioreactor (m2p-labs, Baesweiler, Germany) [46]. MTP-R48-B(OH) round plates were inoculated with 1500 µL culture per well and incubated at 1000 rpm at 28 °C. Backscatter light with a gain of 25 or 20 and Gfp fluorescence (excitation/emission wavelengths: 488/520, gain 80) were used to determine biomass and accumulation of Gfp.

### 2.4. Transcriptional and Post-Translational Regulation of Gus-Cts1 Secretion

To assay regulated secretion, precultures were grown in 5 mL YepsLight for 24 h at 28 °C at 200 rpm. 200 µL culture was transferred into 5 mL fresh YepsLight medium and grown for an additional 8 h under identical conditions. After regeneration, cultures were diluted to reach a final OD_600_ of 1.0 after 16 h in CM-glc or CM-ara. Since *U. maydis* proliferates slower in arabinose, inoculation volume for arabinose cultures was increased by 60%. Cultures were harvested at OD_600_ 0.8 to 1.0 by centrifugation of 2 mL culture at 1500× *g* for 5 min. 1.8 mL supernatants were transferred to fresh reaction tubes and stored at −20 °C until Gus activity determination. 

To assay post-translational regulation, cells were incubated in CM-ara or CM-ara containing 1 µM (f.c.) NA-PP1. Since cultures grow slower when arabinose is used as carbon source and NA-PP1 was added to the medium, the inoculum was increased by 130%. 

For evaluation of time-dependent secretion using both transcriptional and post-translational regulation, strains were inoculated in CM-glc, CM-ara and CM-ara with NA-PP1 to reach a final OD_600_ of 1.0 after 16 h. Cells were then washed in H_2_O and resuspended in CM-ara. Supernatant samples were taken 0, 1, 2, 4, and 8 h post-induction as described above, and Gus activity was determined.

### 2.5. Quantification of Unconventional Secretion Using the Gus Reporter

Extracellular Gus activity was determined to quantify unconventional Cts1 secretion using the specific substrate 4-methylumbelliferyl-β-d-glucuronide (MUG, bioWORLD, Dublin, OH, USA). Cell-free culture supernatants were mixed 1:1 with 2× Gus assay buffer (10 mM sodium phosphate buffer pH 7.0, 28 µM β-mercaptoethanol, 0.8 mM EDTA, 0.0042% (*v/v*) lauroyl-sarcosin, 0.004% (*v/v*) Triton X-100, 2 mM MUG, 0.2 mg/mL (*w*/*v*) BSA) in black 96-well plates. Relative fluorescence units (RFUs) were determined using a plate reader (Tecan, Männedorf, Switzerland) for 100 min at 28 °C with measurements every 5 min (excitation/emission wavelengths: 365/465 nm, Gain 60). For quantification of conversion of MUG to the fluorescent product 4-methylumbelliferone (MU), a calibration curve was determined using 0, 1, 5, 10, 25, 50, 100, 200 µM MU.

### 2.6. SDS PAGE and Western Blot Analysis

To verify protein production and secretion in cell extracts and supernatants, respectively, Western blot analysis was used. 50 mL cultures were grown to an OD_600_ of 1.0 and harvested at 1500× *g* for 5 min in centrifugation tubes. Until further preparation, pellets were stored at −20 °C while supernatants were supplemented with 10% trichloracetic acid (TCA) and incubated on ice. For preparation of cell extracts, cell pellets were resuspended in 1 mL cell extract lysis buffer (100 mM sodium phosphate buffer pH 8.0, 10 mM Tris/HCl pH 8.0, 8 M urea, 1 mM DTT, 1 mM PMSF, 2.5 mM benzamidine, 1 mM pepstatinA, 2× complete protease inhibitor cocktail [Sigma/Aldrich, Billerica, MA, USA]) and agitated with glass beads at 1500 rpm for 10 min at 4 °C. Subsequently, the cell suspension was frozen in liquid nitrogen and crushed in a pebble mill (Retsch, Haan, Germany; 2 min at 30 Hz, 2 times). After centrifugation (6000× *g* for 30 min at 4 °C), the supernatant was separated from cell debris and was transferred to a fresh reaction tube. Protein concentration was determined by Bradford assay (BioRad, Hercules, CA, USA) [47] and 10 µg total protein was used for SDS-PAGE. For the enrichment of proteins from culture supernatants, TCA supplemented supernatants were kept at 4 °C for at least 6 h and centrifuged at 22,000× *g* for 30 min at 4 °C. The precipitated protein pellets were washed twice with −20 °C acetone and resuspended in 3× Laemmli buffer (neutralized with 120 mM NaOH). Samples were boiled at 95 °C for 10 min and centrifuged for 2 min 22,000× *g* prior to application for SDS-PAGE. SDS-PAGE was conducted using 10% (*w*/*v*) acrylamide gels. Subsequently, proteins were transferred to methanol-activated PVDF membranes using semi-dry Western blotting. SHH-tagged Gus-Cts1 was detected using a primary anti-HA antibody (1:4000, Millipore/Sigma, Billerica, MA, USA). For detection of Gfp-tagged proteins such as Don3-Gfp, Don3*-Gfp or Jps1-Gfp a primary anti-Gfp antibody was used (1:4000, Millipore/Sigma, Billerica, MA, USA). An anti-mouse IgG-horseradish peroxidase (HRP) conjugate (1:4000 Promega, Fitchburg, WI, USA) was used as secondary antibody. HRP activity was detected using the Amersham ™ ECL ™ Prime Western Blotting Detection Reagent (GE Healthcare, Chalfont St Giles, UK) and a LAS4000 chemiluminescence imager (GE Healthcare Life Sciences, Freiburg, Germany).

### 2.7. Enzyme-Linked Immunosorbent Assay (ELISA)

For detection of binding activity of respective anti-GfpNB-Cts1 fusions, protein adsorbing 384-well microtiter plates (Nunc^®^ MaxisorpTM, ThermoFisher Scientific, Waltham, MA, USA) were used. Wells were coated with 1 µg Gfp. Recombinant Gfp was produced in *E. coli* and purified by Ni^2+^-chelate affinity chromatography as described earlier [28]. 2 µg BSA dealt as negative control (NEB, Ipswich, MA, USA). Samples were applied in a final volume of 100 µL coating buffer (100 mM Tris-HCL pH 8, 150 mM NaCl, 1 mM EDTA) per well at room temperature for at least 16 h. Blocking was conducted for at least 4 h at room temperature with 5% (*w*/*v*) skimmed milk in coating buffer. Subsequently, 5% skimmed milk in PBS (5% (*w*/*v*) skimmed milk, 137 mM NaCl, 2.7 mM KCl, 10 mM Na_2_HPO_4_, 1.8 mM KH_2_PO_4_, pH 7.2) were added to respective volumes or defined protein amounts of anti-GfpNB-Cts1 samples purified from culture supernatants or cell extracts via Ni^2+^-NTA gravity flow and respective controls. 100 µL of sample were added to wells coated with GFP and BSA. The plate was incubated with samples and controls overnight at 4 °C. After 3× PBS-T (PBS supplemented with 0.05% (*v/v*) Tween-20, 100 µL per well) washing, a mouse anti-HA antibody 1:5000 diluted in PBS supplemented with skimmed milk (5% *w*/*v*) was added (100 µL per well) and incubated for 2 h at room temperature. Then wells were washed again three times with PBS-T (100 µL per well) and incubated with a horse anti-mouse-HRP secondary antibody (50 µL per well) for 1 h at room temperature (1:5000 in PBS supplemented with skimmed milk (5% (*w*/*v*)). Subsequently, wells were washed three times with PBS-T and three times with PBS and incubated with Quanta RedTM enhanced chemifluorescent HRP substrate (50:50:1, 50 µL per well, ThermoFisher Scientific, Waltham, MA, USA) at room temperature for 15 min. The reaction was stopped with 10 µL per well Quanta RedTM stop solution and fluorescence readout was performed at 570 nm excitation and 600 nm emission using an Infinite M200 plate reader (Tecan, Männerdorf, Switzerland). 

### 2.8. Microscopic Analyses

Microscopic analyses were performed with immobilized early-log phase budding cells on agarose patches (3% (*w*/*v*)) using a wide-field microscope setup from Visitron Systems (Munich, Germany), Zeiss (Oberkochen, Germany) Axio Imager M1 equipped with a Spot Pursuit CCD camera (Diagnostic Instruments, Sterling Heights, MI, USA) and the objective lenses Plan Neofluar (40×, NA 1.3), Plan Neofluar (63×, NA 1.25) and Plan Neofluar (100×, NA 1.4). Fluorescent proteins were detected with an HXP metal halide lamp (LEj, Jena, Germany) in combination with filter set for Gfp (ET470/40BP, ET495LP, ET525/50BP). The microscopic system was controlled by the software MetaMorph (Molecular Devices, version 7, Sunnyvale, CA, USA). Image processing including rotating and cropping of images, scaling of brightness, contrast, and fluorescence intensities as well as insertion of scaling bars was performed with MetaMorph. Arrangement and visualization of signals by arrowheads was performed with Canvas 12 (ACD Systems, Victoria, BC, Canada).

## 3. Results 

### 3.1. Evaluating Jps1 as a Regulator for Unconventional Protein Export

The presence of the potential anchoring factor Jps1 is essential for unconventional Cts1 secretion via the fragmentation zone (Figure 1B,C) [30]. This mechanistic insight might provide the unique possibility of using Jps1 as a regulator for unconventional protein secretion and thus, to establish a first inducible system. To test transcriptional induction of Cts1 export via *jps1*, we used derivatives of laboratory strain AB33 lacking the native gene copy of *jps1* and complemented them with P*_crg_* regulated versions of *jps1* or *jps1-gfp,* a fusion to the gene sequence for the green fluorescence protein, encoding a functional fusion protein (Jps1-Gfp; Figure 2A) [30]. Activity of the P*_crg_* promoter depends on the carbon source: The promoter is switched “off” in the presence of glucose and “on” in the presence of arabinose [48]. In addition, the strains carried the established reporter Gus-Cts1 as a read-out for unconventional secretion (Figure 2A) [14]. Microscopic analysis revealed that, as expected, the regulated strains grew yeast-like without any different morphological phenotype both in glucose and in arabinose-containing media. However, in contrast to previous localization studies [30], Jps1-Gfp mainly formed intracellular aggregates (about 80%) during all stages of cytokinesis, with only a minor population of about 3% showing the expected localization in the fragmentation zone in late cytokinesis when transcription was induced by arabinose (Figure 2B and Appendix A). By contrast, control cultures with native Jps1 regulation showed localization in this area in 24% of all investigated cells, likely corresponding to the fraction of cells in the late stage of cytokinesis (Appendix A) [30]. This suggests that deregulation of Jps1 via P*_crg_* interferes with its very specific, cytokinesis-dependent localization. Analysis of unconventional secretion in these strains using the reporter Gus-Cts1 in “off” and “on” conditions revealed that extracellular Gus activities were higher in arabinose than in glucose-containing media, indicating that transcriptional regulation of *jps1* and *jps1-gfp* was successful. However, the base line was elevated and induction levels ranged below two-fold (Figure 2C and Appendix A). Of note, extracellular Gus activity of a control strain with unconventionally secreted Gus-Cts1 in the *jps1* deletion background (lacking regulated *jps1*) was 4.3 times higher when grown in glucose than the activity during growth in arabinose (Figure 2C and Appendix A). This suggests that one reason for the weak induction might be the high background. Additionally, the mislocalization of deregulated Jps1-Gfp likely reduces its function during unconventional secretion suggesting that the lock-type mechanism might not efficiently take place in these conditions. Thus, in the present setup transcriptional regulation via *jps1* is not suitable with respect to biotechnological application for the protein expression platform. 

### 3.2. Transcriptional Regulation of Don3 for Unconventional Protein Export

Since regulation via Jps1 was not convincing for establishing an efficient inducible protein expression system in the present form, we revisited published results on the transcriptional regulation of Don3. Studying the Cts1 export mechanism we had observed that induced *don3* expression via the P*_crg_* promoter reconstitutes unconventional secretion, confirming the lock-type mechanism via the fragmentation zone. The used AB33 Gus-Cts1 derivatives lack the endogenous *don3* copy and were complemented with *don3* or *don3-gfp* regulated by the P*_crg_* promoter (Figure 3A) [31]. We now reproduced these results focusing on the relevant points for biotechnological application. As observed earlier, the aggregation phenotype was complemented and Don3-Gfp localized to fragmentation zones in arabinose-containing medium (Appendix A) [31,32]. Deregulated Don3-Gfp solely localized to fragmentation zones of dividing cells (Appendix A) which is identical to published results [49]. Reporter assays revealed induction levels of extracellular Gus activity ranging between five- and seven-fold for Don3-Gfp and Don3, respectively, indicating efficient transcriptional regulation (Appendix A). In Western blot analyses, Don3-Gfp was detected as a full-length protein in cell extracts of cultures grown in arabinose. Culture supernatants revealed the presence of free Gfp, suggesting that the full-length protein is secreted into the extracellular space where Don3 is quickly degraded (Figure 3B; Appendix A). This is likely caused by secreted proteases, a well-known phenomenon in fungi including *U. maydis* [15,31]. The high stability of the remaining Gfp is presumably due to its robust beta-barrel structure [50]. A control strain for arabinose induction and cell lysis carrying the gene sequence for cytosolic Gfp under control of P*_crg_* was used as a control (AB33 P_crg_gfp) and revealed the presence of cytosolic Gfp in cell extracts but not in culture supernatants of cultures grown in arabinose (Figure 3B). 

The results confirmed that Don3-mediated regulated secretion efficiently separates cell growth and protein synthesis from secretion. Heterologous proteins are thus kept protected in the cell prior to secretion. For transformation of our findings into a biotechnological process, cycles between cell growth, induction, and protein harvest would be useful. This, for example, reduces the exposure time of the secreted heterologous product in the culture supernatant and thus, potential proteolytic degradation. We tested the robustness of such a strategy by switching between “on” and “off” conditions in cycles over five days while tracking unconventional secretion via the Gus-Cts1 reporter. Indeed, the complete process was reversible and induction levels were comparable throughout the different cycles (Figure 3C). This suggests that transcriptional regulation of *don3* is a valuable new tool for heterologous protein production in a cyclic process.

In summary, we established a first regulatory strategy for unconventional protein export using a nutrient-dependent promoter.

### 3.3. Post-Translational Regulation of Don3 for Unconventional Protein Export

Diauxic switches of the carbon source are associated with severe changes in the metabolism of the cell [51] and may thus also influence protein production. Therefore, we aimed to test an additional method based on chemical genetics to regulate unconventional secretion without causing a strong metabolic burden to the cell. It is well established that bulky ATP analogs in concert with mutagenized kinase versions can be used to inactivate protein kinases [52]. This has also been shown for Don3 using the ATP analogue NA-PP1 (1-(1,1-dimethylethyl)-3-(1-naphthalenyl)-1H-pyrazolo[3,4-d]pyrimidin-4-amine) in previous studies [49,53]. Based on this, we tested, if post-translational regulation of Don3 activity could be used to regulate Cts1-mediated unconventional secretion. Thus, we adapted our regulated system and introduced a respective amino acid exchange in Don3 (M157A) which allows acceptance of the reversible inhibitor (Don3*; Figure 4A). When cells were grown under promoter “on” conditions in arabinose with the ATP analog, we observed cell aggregates, suggesting that inhibition of kinase activity was successful, while arabinose cultures lacking the analog grew normal (Figure 4B). Accordingly, Don3*-Gfp accumulated at the primary septum of cell aggregates in the presence of the ATP analog (Figure 4B), suggesting that the mutation disrupts kinase activity but does not impair biosynthesis and localization of the protein. By contrast, in cells grown without the analog, Don3*-Gfp fluorescence was observed at mother-cell boundaries of budding cells, resembling the natural situation (Figure 4B) [31]. On the level of unconventional Cts1 secretion, we observed diminished extracellular Gus activity in the presence of the ATP analog and about a five-fold increase in activity in its absence for regulated Don3*-Gfp and seven-fold for regulated Don3* (Figure 4C). Western blot analyses confirmed that Don3*-Gfp was present in cell extracts independently from addition of NA-PP1 while free Gfp was only present in culture supernatants grown without the bulky analog. This suggests that Don3*-Gfp is unconventionally secreted only under these conditions (Figure 4D and Appendix A). These results confirm that post-translational regulation of Don3* is a second possibility to create a regulatory switch, providing the advantage of minimal invasiveness. Thus, we succeeded in establishing a tailor-made strategy to regulated unconventional secretion without drastic metabolic impact for the production host due to adaptation to new media. However, absolute induction levels were slightly lower than for transcriptional regulation (Appendix A).

### 3.4. Time-Resolved Comparison of Regulatory Switches

To further elucidate the effects of transcriptional and post-transcriptional Don3 regulation, we directly compared both regulatory methods in a time-resolved manner. Therefore, the strain expressing Don3* was grown in three different media overnight: (i) arabinose for constitutive unconventional secretion, (ii) glucose for transcriptional inhibition of unconventional secretion, and (iii) arabinose and kinase inhibitor NA-PP1 for post-translational inhibition of unconventional secretion (Figure 5). Subsequently, cells were washed to remove media components including all previously exported Gus-Cts1 and resuspended in fresh medium containing only arabinose without NA-PP1 for constitutive induction of unconventional secretion. Gus activities were determined after induction at distinct time points for eight hours (Figure 5). Cultures pre-grown in glucose showed a high level of induction two hours after medium switch (light blue columns), suggesting that cell aggregates had resolved and accumulated Gus-Cts1 had been secreted at this time point. By contrast, cultures pre-grown with arabinose and the inhibitor for post-translational induction reached similar levels already one hour post-induction (dark blue columns). This is likely due to the fact, that inactive Don3* is produced and localized to the mother-cell boundary in these cells already during the pre-incubation overnight. After removal of the inhibitor, the protein can directly fulfill its function in secondary septum assembly, while after transcriptional inhibition, both the transcript and the resulting translation product first need to be synthesized. The quick response after release of post-translational inhibition is in accordance with earlier studies where kinase inhibition by NA-PP1 was used to address the function of Don3 during septation [49,53]. By comparison, cultures grown overnight under constitutive induction in arabinose had no intracellular storage of Don3* due to its unconventional secretion during cell separation [31]. These cultures thus showed only very weak extracellular Gus activities in the first few hours after induction (white columns). They reached a comparable level to the other cultures only after four hours. After 8 hours, all cultures exhibited extracellular Gus activities, which were not significantly different from each other anymore. The difference in immediate induction levels between the culture preincubated in arabinose lacking NA-PP1 and those preincubated in glucose or arabinose with NA-PP1 might be further boosted by the fact that the latter cells are present in aggregates prior to induction and all these start budding at the same time after induction. These data demonstrate that both regulated systems are advantageous compared to constitutive secretion when cell harvest is conducted within the first few hours after induction.

In summary, regulation was successfully achieved on two different levels, namely exploiting transcriptional and post-translational induction of the gene expression and gene product activity of septation factor Don3, respectively.

### 3.5. Establishing an Autoinduction Process Based on Transcriptional Regulation

The previously established regulatory tools for Don3 depend on medium switches, which are not easily compatible with biotechnological processes, especially during upscaling in a bioreactor. Hence, we tested if autoinduction can be used to avoid the medium switch but keep the advantage of separated growth/protein synthesis and secretion phase. To establish such a process, we concentrated on transcriptional regulation as anf inexpensive tool. We assayed the activity of the P*_crg_* promoter in the presence of different concentrations of glucose and arabinose resembling “off” and “on” state of the system, respectively, using an AB33don3Δ derivative expressing *gfp* under control of the arabinose inducible P*_crg_* promoter as a transcriptional reporter (AB33don3Δ/P_crg_gfp). The resulting Gfp protein accumulates in the cytoplasm and can easily be detected by its fluorescence. The strain was cultivated in a BioLector device with online monitoring of Gfp fluorescence and scattered light as a read-out for fungal biomass in minimal volumes (Figure 6 and Appendix A) [46]. Initially, either 1% glucose (56 mM) or 1% arabinose (67 mM) or the two sugars in different ratios were used (0.25:0.75%/0.5:0.5%/0.75:0.25%, adding up to 1% each; Appendix A). The Gfp read-out indicated a steadily rising Gfp signal for cultures growing in arabinose as the sole carbon source, while cultures growing in glucose showed only weak background fluorescence. In the presence of different ratios of mixed glucose and arabinose, cultures consumed the preferred carbon-source glucose first and switched to arabinose later, presumably, when the respective amount of glucose was completely metabolized. In general, during cultivation in glucose and arabinose, Gfp fluorescence remained very low during consumption of glucose, followed by an increasing Gfp fluorescence after switching to arabinose (Appendix A). The prolonged phase with low fluorescence at a time when biomass is already constantly increasing is a prerequisite for successful autoinduction and indicated that the strategy is successful.

Next, to identify the optimal composition for an autoinduction medium, which is characterized by a prolonged growth phase with minor promoter activity in the beginning and a high plateau of Gfp fluorescence after induction (i.e., after consumption of glucose), we varied the total sugar amounts and the ratios of glucose and arabinose in the medium (Figure 6A,B). Again, cultures containing only 1% arabinose or glucose were used as controls (light gray dots, light green lines). For two other cultures, initial biomass formation was initiated with 1% glucose, while induction of the P*_crg_* promoter and thus *gfp* expression after glucose consumption was stimulated by either 1% or 2% arabinose. Compared to the arabinose control, these cultures showed a delayed accumulation of Gfp fluorescence indicating successful uncoupling of growth and protein production. The total Gfp fluorescence was more than two-fold higher with elevated total sugar concentrations, which is in line with a higher total biomass (Figure 6A,B). However, interestingly, higher initial glucose concentrations of two or three percent did not result in higher biomass formation or Gfp yield (Appendix A). A possible explanation is that other factors besides the carbon source in the medium become limiting. The increase of arabinose from 1% to 2% did yield higher biomass but no further increase in Gfp fluorescence (Figure 6A,B). Therefore, medium containing 1% glucose and 1% arabinose was selected for further autoinduction experiments (see below).

In summary, we established a simple autoinduction protocol that can be applied in a broad variety of biotechnological processes without the need for medium switches.

### 3.6. Applying Autoinduction for the Export of Functional Nanobodies

Finally, we applied autoinduction via transcriptional *don3* regulation for the unconventional secretion of heterologous proteins using a nanobody as an example for an established pharmaceutical target protein [54]. Therefore, we generated a strain in which a fusion of an anti-Gfp nanobody [55] with Cts1 (NB-Cts1) as carrier was expressed by the previously established strategy (AB33don3Δ/P_crg_don3/NB-Cts1; Figure 7A). Unconventional secretion of the functional nanobody using Cts1 as a carrier had been established in an earlier study [28]. Western blot analysis verified the production and secretion of the fusion protein in arabinose medium (Appendix A). Next, we cultivated the strain in buffered autoinduction medium using the most efficient composition (1% glucose, 1% arabinose) in shake flasks and followed synthesis of functional NB-Cts1 fusion protein along the cultivation by BioLector online monitoring (Figure 7B) in concert with enzyme-linked immunosorbent assays (ELISA) using the cognate antigen Gfp (Figure 7C,D and Appendix A). Gfp binding activity was barely detectable after 8 h of incubation in autoinduction medium in purified culture supernatants (Figure 7C) but clearly in cell extracts (Figure 7D). After 15 h, ELISA values were strongly enhanced for NB-Cts1 purified from culture supernatants (Figure 7B). This corresponded to the time when glucose was presumably depleted from the medium. These results are consistent with the parallel evaluation of an identical culture in the BioLector device. Here, the diauxic switch caused clear adaptations in pH and Dissolved Oxygen Tension (DOT) after approximately 15 h of cultivation (Figure 7B). Thus, an efficient autoinduction process was established on the basis of transcriptional *don3* regulation, allowing for the production of functional heterologous proteins by unconventional secretion.

In essence, we successfully applied regulated unconventional secretion for the export of nanobodies.

## 4. Discussion

In this study, we build on our mechanistic knowledge on Cts1 export to establish the first regulatory systems to control the unconventional secretion of heterologous proteins in *U. maydis*. Systems for regulated or inducible protein production are widespread within the different expression systems. They enable a strict temporal control of the protein production process because growth and protein synthesis can be largely separated [56]. Although regulated systems are well established for protein production, they are usually based on the direct transcriptional regulation of the promoter of the gene-of-interest [57]. Here, we went one step beyond and regulated the mechanism of secretion rather than the gene-of-interest itself. While a deep knowledge of the conventional secretion pathway in eukaryotes exists [58], we are not aware of any regulatory system based on these mechanistic insights that are currently applied for heterologous protein production, at least in fungal systems.

Regulation based on septation factor kinase Don3 was successfully achieved on two different levels, namely exploiting transcriptional and post-translational inhibition of the gene expression and gene product activity, respectively. Don3 is essential for secondary septum formation [32] and thus acts as a kind of gate keeper for lock-type unconventional secretion [29,30,31]. When Don3 is not present or inactive, the product destined for Cts1-mediated secretion is formed along with the cell growth but trapped within the cell where it is protected from frequently occurring extracellular proteases [15,27]. Both regulatory levels are powerful tools for biotechnological application: while transcriptional control is inexpensive and useful for cheap products, post-translational control is more expensive due to the need of inhibitor but comes with a faster release of the protein avoiding long exposure of the product to proteases. Thus, the latter method is appropriate for high prize products such as pharmaceutical proteins exemplified by antibody formats such as nanobodies [59]. Furthermore, proteins that are prone to proteolytic degradation might benefit from the fast release by post-translational induction.

By contrast, only a weak induction was achieved using transcriptional *jps1* regulation. Controlling Jps1 expression would be very attractive because its absence is not connected to morphological changes as observed for Don3 [30]. In the future, the use of alternative inducible promoters might lead to a significant improvement of the system by reducing the background activity during “off” conditions. For example, an orthogonal system such as tetracycline-regulated gene expression could be used [60]. It avoids metabolic effects that might arise with nutrient-dependent promoters and allows for titration of the expression strength. The system has already been applied in fungi including *U. maydis* [61,62] but needs careful adaptation to the respective application.

Using the example of transcriptional *don3* control in combination with an autoinduction protocol resulted in a first bioprocess. Optimization of yield and simplification of the experimental procedures by reduction of user intervention after culture inoculation are major advantages associated with autoinduction processes applied in industrial biotechnology. While lactose-derived autoinduction is applied in *E. coli* for the T7lac promoter system for years [63,64], glycerol/methanol-based autoinduction of the *AOX1* promoter was recently also described for *P. pastoris* as a fungal model organism [65]. Here, we add a protocol for autoinduction of unconventional secretion to the list.

In the future, we might further adopt our system and establish sophisticated optogenetic regulation [66,67]. Light-dependent transcriptional regulation can for example be achieved using phytochromes [68,69]. One elegant example for regulation of protein stability is the use of photosensitive degrons derived from plant proteins, which are already successfully applied in *S. cerevisiae* [70,71]. The advantage of such systems is that they are non-invasive and allow for a precise temporal control of the induction process [66,67]. In summary, we here substantially improved the method portfolio for our unconventional protein secretion system and went a further step ahead towards a novel fungal expression platform.

## Figures and Tables

**Figure 1 jof-07-00179-f001:**
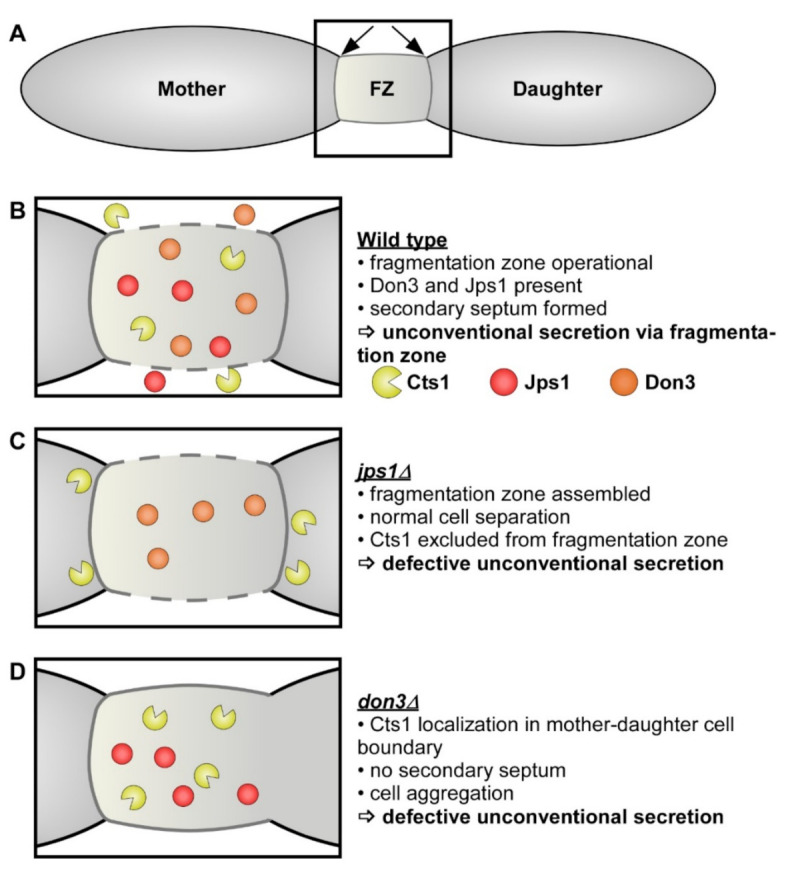
Current schematic model of lock-type secretion and implications for heterologous protein export in *U. maydis*. (**A**) Unconventional secretion of chitinase Cts1 occurs during cytokinesis of yeast cells. Prior to budding, a primary septum is assembled at the mother cell side, followed by a secondary septum at the daughter cell side. The two septa delimit a so-called fragmentation zone (FZ), a small compartment filled with different proteins and membrane vesicles (not shown). Position of septa is indicated by arrows. (**B**) In the wild-type situation, Cts1 accumulates in the fragmentation zone and participates in cell separation. Recent research identified the potential anchoring factor Jps1 and the septation factors Don1 (not shown) and Don3, which are essential for Cts1 secretion. (**C**) In the absence of Jps1, Cts1 is excluded from the fragmentation zone and unconventional secretion is abolished. Nevertheless, cell separation occurs normally. (**D**) In the absence of Don3, the secondary septum is not assembled, and cell separation is hampered, leading to the formation of cell aggregates. Cts1 still accumulates at the mother-daughter cell boundary but its unconventional secretion is abolished.

**Figure 2 jof-07-00179-f002:**
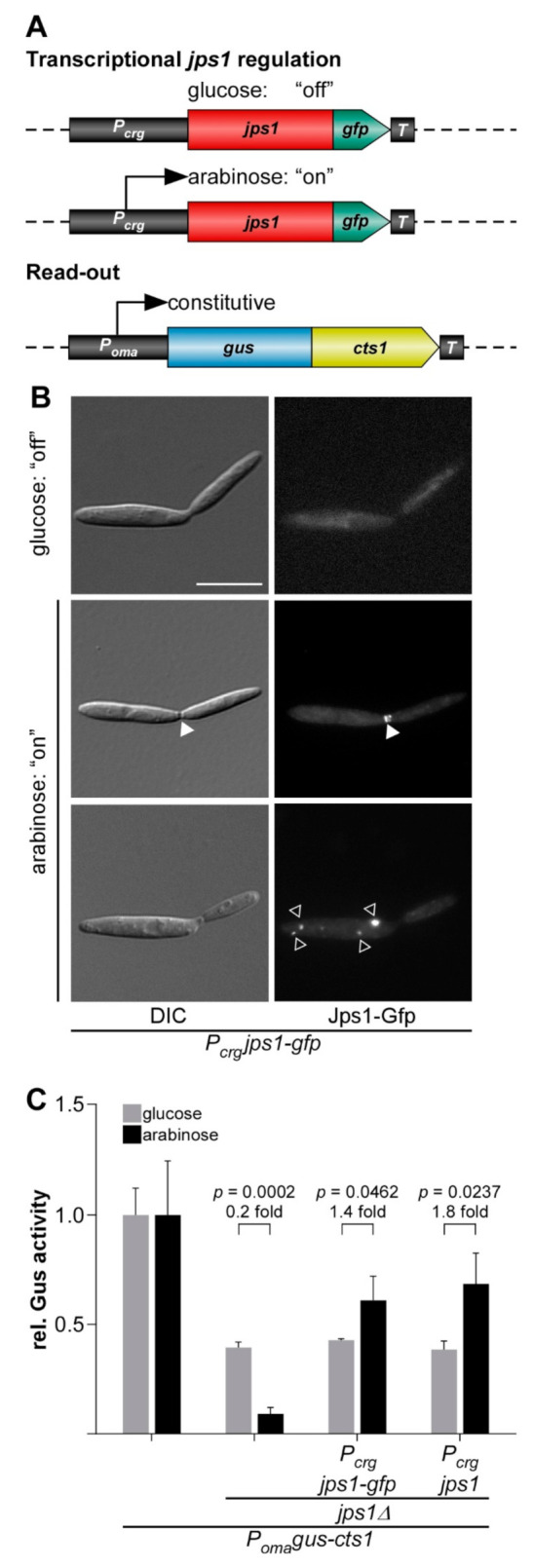
Transcriptional regulation of unconventional secretion via the potential anchoring factor Jps1. (**A**) Rationale of regulated Jps1 expression on the genetic level. Unconventional secretion factor Jps1 is controlled by the arabinose inducible promoter P*_crg_* and constitutively produced Gus-Cts1 is used as a read-out for quantification of unconventional secretion. *T*, transcriptional terminator. (**B**) Micrographs of yeast-like growing cells in the “on” and “off” stage mediated by glucose and arabinose in the medium, respectively. White arrowheads depict the fragmentation zone between mother-daughter cell boundary, open white arrowheads show additional intracellular accumulations of Jps1-Gfp. DIC, differential interference contrast. Scale bar, 10 µm. (**C**) Gus activity in culture supernatants of indicated AB33 Gus-Cts1 derivatives. Enzymatic activity was individually normalized to average values of positive controls secreting Gus-Cts1 constitutively, which were grown in glucose and arabinose-containing cultures. Values for the positive control in the two media do not differ significantly (*p* = 0.2022; Appendix A). Strains containing regulated *jps1* or *jps1-gfp* versions show a slight induction of extracellular Gus activity after growth in arabinose-containing medium. Error bars depict standard deviation. The diagram represents results of three biological replicates. Fold change of induced cultures and *p*-values of Student’s unpaired *t*-test are shown. Definition of statistical significance: *p*-value < 0.05.

**Figure 3 jof-07-00179-f003:**
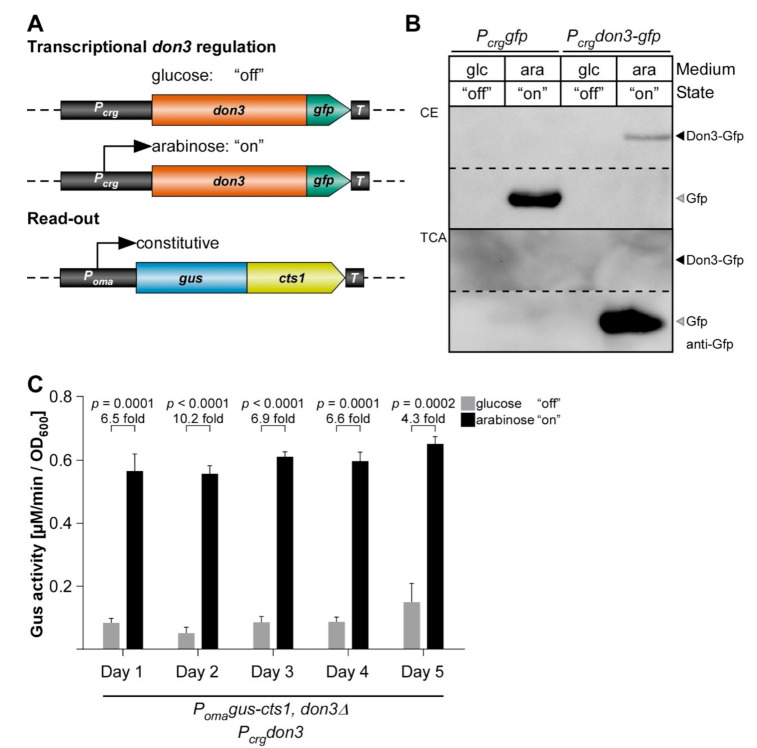
Transcriptional regulation of unconventional secretion via kinase Don3. (**A**) Exemplary strategy for transcriptional *don3-gfp* regulation of unconventional secretion. Upon supplementation of the medium with glucose, the P*_crg_* promoter is inactive, while the addition of arabinose leads to its activation. Constitutive Gus-Cts1 expression is used as a read-out for quantification of unconventional secretion. (**B**) Western blot of cell extracts (CE, upper panel) and TCA precipitated culture supernatants (TCA) depicting Don3-Gfp and cytosolic Gfp (cell lysis control). Primary antibodies against Gfp were used to detect the respective proteins (anti-Gfp). In cell extracts, Don3-Gfp protein is only present upon induction with arabinose. Glc, glucose supplementation; ara, arabinose supplementation. (**C**) Induction of unconventional secretion is reversible upon shift between glucose and arabinose supplementation using strain AB33don3Δ/P_crg_don3/Gus-Cts1. Cultivation of cells in cycles consisting of 16-h growth in CM-glc (supplemented with glucose) and 8 h CM-ara (supplemented with arabinose), allows alternating “on” and “off” states of unconventional secretion. After each cycle, the relative extracellular activity of Gus-Cts1 was determined and cell densities were adjusted for the next cycle. The experiment was conducted over 5 consecutive days. Error bars depict standard deviation. The diagram represents results of four biological replicates. Fold change of induced cultures and *p*-values of Student’s unpaired *t*-test are shown. Definition of statistical significance: *p*-value < 0.05.

**Figure 4 jof-07-00179-f004:**
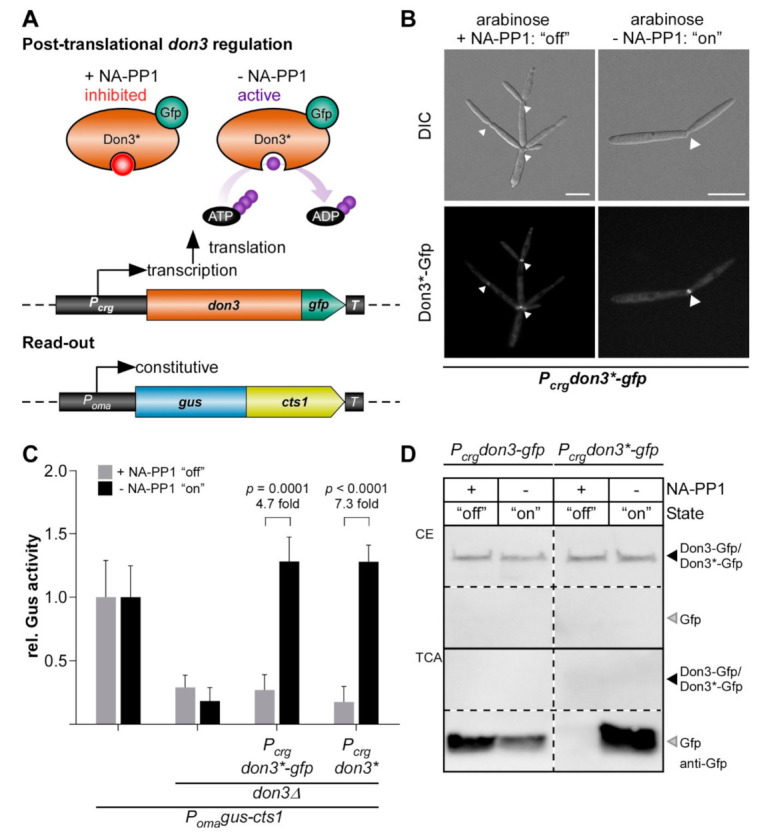
Post-translational regulation of unconventional secretion via inactivation of Don3 kinase activity. (**A**) Strategy for post-translational regulation of unconventional secretion using the mutagenized Don3 version Don3* in concert with a bulky ATP analog (NA-PP1). (**B**) Micrographs of yeast-like growing cells grown in medium containing arabinose. Cells treated with the bulky ATP analog NA-PP1 are indicated. Arrowheads depict the Gfp signal at the mother-daughter cell boundary. DIC, differential interference contrast. Scale bars, 10 µm. (**C**) Gus activity in culture supernatants of indicated AB33 P_oma_Gus-Cts1 derivatives. Enzymatic activity was individually normalized to average values of positive controls secreting Gus-Cts1 constitutively, which were grown in arabinose-containing cultures. Values of positive controls in the two media do not differ significantly (*p* = 0.7317; Appendix A). Strains containing regulated *don3** or *don3*-gfp* versions show a strong induction of extracellular Gus activity after growth in arabinose medium without NA-PP1. The diagram represents results of four biological replicates. Error bars depict standard deviation. Fold change of induced cultures and *p*-values of Student’s unpaired *t*-test are shown. Definition of statistical significance: *p*-value < 0.05. (**D**) Western blots of cell extracts (CE, upper panel) and TCA precipitated culture supernatants of AB33don3Δ cultures expressing regulated Don3-Gfp and Don3*-Gfp. Primary antibodies against Gfp were used to detect the respective proteins (anti-Gfp). Both fusion proteins are degraded in the supernatant and only free Gfp can be detected. Free Gfp derived from Don3-Gfp was detected in the presence and absence of NA-PP1. However, no free Gfp derived from Don3*-Gfp was detectable in the presence of NA-PP1, indicating inhibition of unconventional secretion.

**Figure 5 jof-07-00179-f005:**
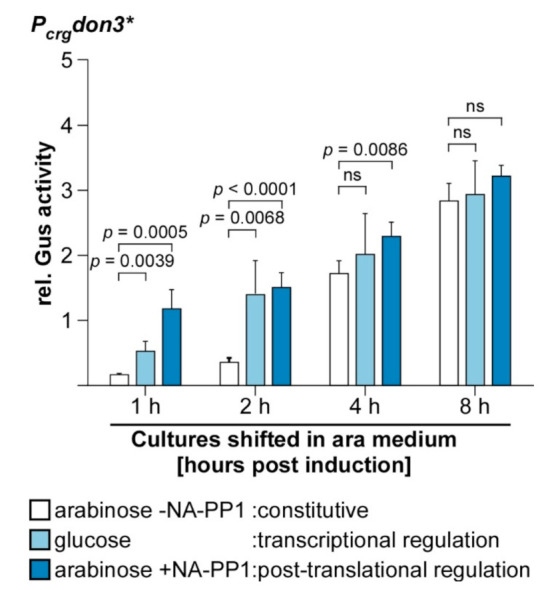
Time-resolved comparison between transcriptional and translational Don3 regulation. Cells of the Gus-Cts1 reporter strain containing the mutagenized kinase version Don3* (Figure 4A) were pre-incubated in medium supplemented with arabinose only (arabinose − NA-PP1, white columns), with glucose only (light blue columns) or with arabinose and the kinase inhibitor NA-PP1 (arabinose + NA-PP1, dark blue columns). After a washing step to remove media components, cells were resuspended in medium containing arabinose and Gus activity was determined for 8 h at distinct time points. Enzymatic activity was normalized to average values of induced overnight culture. The diagram represents results of four biological replicates. Error bars depict standard deviation. *p*-values of Student’s unpaired *t*-test between previously normalized culture and induced culture are shown. Definition of statistical significance: *p*-value < 0.05.

**Figure 6 jof-07-00179-f006:**
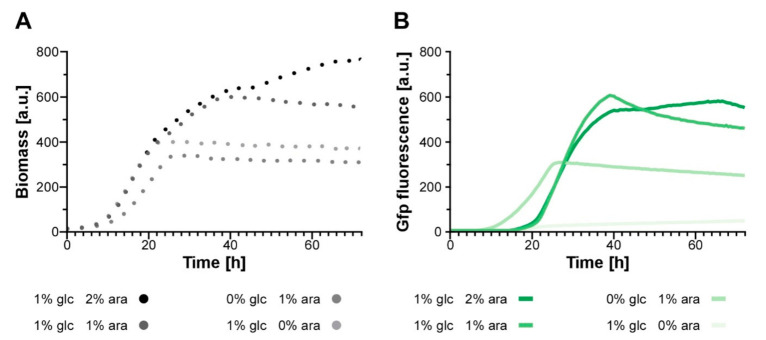
Establishing an autoinduction process based on transcriptional regulation. (**A**,**B**) Reporter strain AB33don3Δ/P_crg_gfp was cultivated in buffered CM medium supplemented with glucose (glc) and arabinose (ara) in different amounts and ratios as indicated in the diagram. Since in contrast to the experiments before the cultures were incubated for a prolonged time reaching high optical densities, the medium was buffered with 100 mM MES to prevent a drastic pH drop [15]. The two parameters fungal biomass (**A**) and Gfp fluorescence (**B**) were recorded online in a BioLector device. Gains: 20 (scattered light); 80 (Gfp).

**Figure 7 jof-07-00179-f007:**
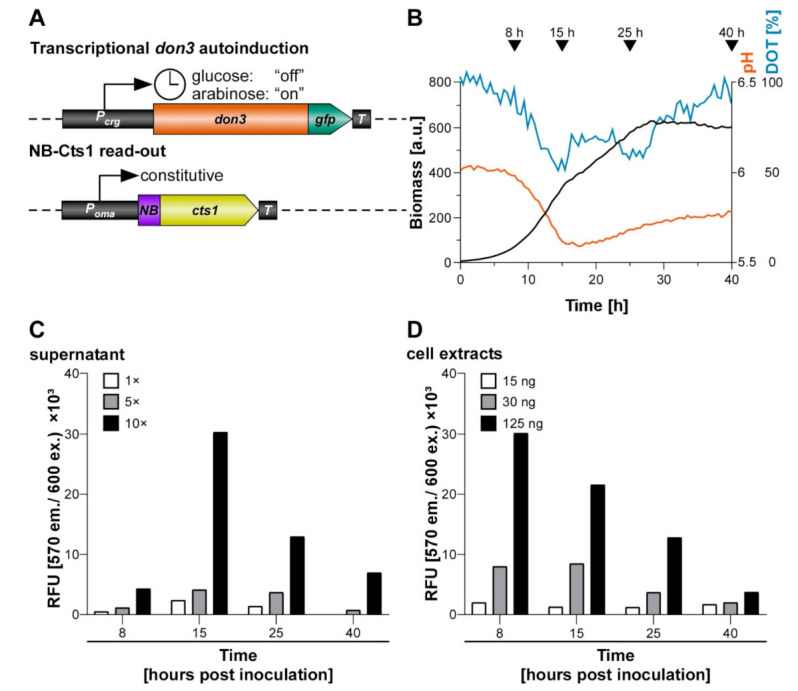
Evaluation of the autoinduction process for unconventional secretion of an anti-Gfp nanobody. Strain AB33don3Δ/P_crg_don3/NB-Cts1 was inoculated in CM medium supplemented with 1% glucose, 1% arabinose, and buffered with 0.1 M MES. The culture was split into 5 individual flasks for harvest of supernatant proteins, cell extracts and parallel online growth monitoring in BioLector and offline monitoring via photometer. Supernatant was collected at defined time points and unconventionally secreted NB-Cts1 was IMAC purified. Cell extracts were prepared in parallel. For purified supernatant and cell extracts ELISA were performed using purified Gfp as antigen. (**A**) Schematic representation of the genetic setup for transcriptional *don3-gfp* regulation of unconventional secretion in autoinduction medium by activating transcription through arabinose after the consumption of glucose (diauxic switch indicated by clock symbol). NB-Cts1 is constitutively produced but trapped in the cell prior to Don3 synthesis. (**B**) Online monitoring of the cultivation using the BioLector device. Primary ordinate axis shows biomass via backscatter light (gain 20), secondary ordinate axis shows pH, red, and dissolved oxygen tension (DOT), blue. Time points of sampling of parallel grown shake flask cultures are indicated by arrowheads. (**C**) Enzyme-linked immunosorbent assay (ELISA) using NB-Cts1 purified from culture supernatants at indicated time points. 1×, 5× and 10× concentrated purified supernatants, (**D**) ELISA using cell extracts harvested at defined protein amounts containing 15 ng, 30 ng, and 125 ng total protein.

**Table 1 jof-07-00179-t001:** DNA oligonucleotides used in this study.

Designation	Nucleotide Sequence (5′–3′)
oUM910	GATCCAATTGATGCCAGGCATCTCCAAGAAGCC
oUM911	GATCGGCGCGCCTTAGGATTCCGCATCGATTGGGG
oUM912	GATCGGCGCGCCTTACTTGTACAGCTCGTCCATGC
oAB23	GCTACAAGCTCTGGATCATTGCTGAGTATCTAGCAGGTGGATCC
oAB24	GGATCCACCTGCTAGATACTCAGCAATGATCCAGAGCTTGTAGC
oRL946	CCGATCCACAAGCTTCGGTGCTTGGATTGG
oRL947	CGGTGTTGCCATGAACACCGATGGCCAGTG
oRL948	GGTACTTGTGCTCGGGGAACACCTCGGCGA
oRL949	GTTTTGTCTCGTTCCGTGCGTCGACGACAGA
oMF502	ACGACGTTGTAAAACGACGGCCAG
oMF503	TTCACACAGGAAACAGCTATGACC
oUP65	GGAATTCCATATGGCGAGCCTTGAGGCTGCGTTCC
oUP66	CGGGATCCGATTTGCAAGTCGTGGGCCTTCG
oMB190	GATTACAGGATCCATGCCAGGCATCTCC
oMB520	CATGAATTCGGATTCCGCATCGATTGGGG
oMB521	TCAGAATTCATGGTGAGCAAGGGCGAGG
oMB522	CATGCGGCCGCCTTACTTGTACAGCTCGTCC

**Table 2 jof-07-00179-t002:** *U. maydis* strains used in this study. Strains were obtained by homologous recombination using antibiotic resistance cassettes for selection: PhleoR, phleomycin resistance; CbxR, carboxin resistance; HygR, hygromycin resistance; NatR, nourseothricin resistance. Don3*, version of kinase Don3 carrying an amino acid exchange at position 157 (methionine replaced by alanine).

Strains	Relevant Genotype/Resistance	Strain Collection No. (Uma ^1^)	Plasmids Transformed/Resistance ^2^	Manipulated Locus	Progenitor (Uma ^1^)	Reference
AB33	*a2 P_nar_bW2bE1*	133	pAB33	*b*	FB2 [43]	[40]
	*PhleoR*					
AB33	*a2 P_nar_bW2bE1* PhleoR	1289	pUMa2113/ CbxR	*ip*	133	[27]
Gus-Cts1	*ip^S^[P_oma_gus:shh:cts1]ip^R^* CbxR					
AB33	*a2 P_nar_bW2bE1* PhleoR	1742	pUMa2717/ HygR	*umag_05543* ^3^ *(don3)*	1289	[31]
don3Δ/	*ip^S^[P_oma_gus:shh:cts1]ip^R^* CbxR					
Gus-Cts1	*umag_05543Δ*_HygR					
AB33	*a2 P_nar_bW2bE1* PhleoR	2028	pUMa2717/ HygR	*umag_05543* *(don3)*	133	[31]
don3Δ	*umag_05543Δ*_HygR					
AB33	*a2 P_nar_bW2bE1* PhleoR	2300	pUMa3328/ NatR	*umag_02178* *(upp1)*	1742	[31]
don3Δ/	*ip^S^[P_oma_gus:shh:cts1]ip^R^* CbxR					
P_otef_ gfp/	*umag_05543Δ*_HygR					
Gus-Cts1	*upp1::[P_otef_gfp]* NatR					
AB33	*a2 P_nar_bW2bE1* PhleoR	2301	pUMa3329/ NatR	*umag_02178* *(upp1)*	1742	This study
don3Δ/	*ip^S^[P_oma_gus:shh:cts1]ip^R^* CbxR					
P_crg_ gfp/	*umag_05543Δ*_HygR					
Gus-Cts1	*upp1::[P_crg_gfp]* NatR					
AB33	*a2 P_nar_bW2bE1* PhleoR	2302	pUMa3330/ NatR	*umag_02178* *(upp1)*	1742	[31]
don3Δ/	*ip^S^[P_oma_gus:shh:cts1]ip^R^* CbxR					
P_crg_don3-gfp/	*umag_05543Δ*_HygR					
Gus-Cts1	*upp1::[P_crg_don3:gfp]* NatR					
AB33	*a2 P_nar_bW2bE1* PhleoR	2303	pUMa3331/ NatR	*umag_02178* *(upp1)*	1742	[31]
don3Δ/	*ip^S^[P_oma_gus:shh:cts1]ip^R^* CbxR					
P_crg_don3/	*umag_05543Δ*_HygR					
Gus-Cts1	*upp1::[P_crg_don3]* NatR					
AB33	*a2 P_nar_bW2bE1* PhleoR	2092	pUMa2775/ HygR	*umag_03776* *(jps1)*	133	[30]
jps1Δ	*umag_03776Δ*_HygR					
AB33	*a2 P_nar_bW2bE1* PhleoR	2991	pUMa2113/ CbxR	*ip*	2092	This study
jps1Δ/	*umag_03776Δ*_HygR					
Gus-Cts1	*ip^S^[P_oma_gus:shh:cts1]ip^R^* CbxR					
AB33	*a2 P_nar_bW2bE1* PhleoR	3053	pUMa4234/ NatR	*umag_02178* *(upp1)*	2991	This study
jps1Δ/	*umag_03776Δ*_HygR					
P_crg_jps1-gfp/	*ip^S^[P_oma_gus:shh:cts1]ip^R^* CbxR					
Gus-Cts1	*upp1::[P_crg_jps1:gfp]* NatR					
AB33	*a2 P_nar_bW2bE1* PhleoR	3054	pUMa4235/ NatR	*umag_02178* *(upp1)*	2991	This study
jps1Δ/	*umag_03776Δ*_HygR					
P_crg_jps1/	*ip^S^[P_oma_gus:shh:cts1]ip^R^* CbxR					
Gus-Cts1	*upp1::[P_crg_jps1]* NatR					
AB33	*a2 P_nar_bW2bE1* PhleoR	3069	pUMa4313/ NatR	*umag_02178* *(upp1)*	1742	This study
don3Δ/	*ip^S^[P_oma_gus:shh:cts1]ip^R^* CbxR					
P_crg_don3*-gfp/	*umag_05543Δ*_HygR					
Gus-Cts1	*upp1::[P_crg_don3^M157A^:gfp]* NatR					
AB33	*a2 P_nar_bW2bE1* PhleoR	3070	pUMa4308/ NatR	*umag_02178* *(upp1)*	1742	This study
don3Δ/	*ip^S^[P_oma_gus:shh:cts1]ip^R^* CbxR					
P_crg_don3*/	*umag_05543Δ*_HygR					
Gus-Cts1	*upp1::[P_crg_don3^M157A^]* NatR					
AB33	*a2 P_nar_bW2bE1* PhleoR	3346	pUMa3331/ NatR	*umag_02178* *(upp1)*	2028	This study
don3Δ/	*umag_05543Δ*_HygR					
P_crg_don3*	*upp1::[P_crg_don3^M157A^]* NatR					
AB33	*a2 P_nar_bW2bE1* PhleoR	3410	pUMa2240/ CbxR	*ip*	3346	This study
don3Δ/	*ip^S^[P_oma_his:*anti-*GfpNB:ha:cts1]ip^R^* CbxR					
P_crg_don3/	*umag_05543Δ*_HygR					
NB-Cts1	*upp1::[P_crg_don3^M157A^]* NatR					
AB33	*a2 P_nar_bW2bE1*PhleoR	2274	pUMa3293/ CbxR	*ip*	2092	This study, Appendix A
jps1Δ	*umag_03776Δ*_HygR					
P_jps1_jps1-gfp	*ip^S^[P_jps1_jps1:gfp]ip^R^*CbxR					

^1^ Internal strain collection numbers. Strains are called UMa plus a 4-digit number as identifier. ^2^ Plasmids generated in our working group are integrated in a plasmid collection and termed pUMa plus a 4-digit number as identifier. ^3^ Genes of *U. maydis* are indicated with a 5-digit *umag* number referring to the current genome annotation at EnsembleFungi [37].

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
