# Peer review of "Controlling Unconventional Secretion for Production of Heterologous Proteins in Ustilago maydis through Transcriptional Regulation and Chemical Inhibition of the Kinase Don3"

_jof, 2021, doi:10.3390/jof7030179_

Round 1
Reviewer 1 Report
Comments for authors.
This manuscript attempts to develop a regulatory system for unconventional protein secretion, using U. maydis as a model organism. The authors utilized the lock-type mechanism during yeast bud separation to control protein export through the regulation of the septation factor Don3, which is required for Cts1 secretion. The heterologous protein can then be fused with Cts1 for production and export to culture media, which reduce costs and complexity of further protein utilization.
Overall, I think this study is really fascinating. The authors utilize basic knowledge of Fungal Cell Biology to develop a novel method for producing secreted proteins without perturbation from conventional protein secretion. The study design is elaborated, and the writing (esp. Results and Discussion) is easy to understand the flow of the study. Thus, I think this manuscript is worth considering for publication.
I have several comments for authors that may improve the quality of their manuscript. As the manuscript file I received did not indicate line numbers, I would provide comments by sections.
Title: I think it would be useful to indicate that the authors work with U. maydis. This is to help audience have an idea about the scope of the paper. It also brings an attention that this unique system in U. maydis is utilizable while other eukaryotic systems are still not due to obscure knowledge.
Abstract: I would recommend mentioning Don3 as it is a key player for controlling protein export in this study. Brief explanation, like saying ‘Regulation of Don3 can delay the formation of fragmentation zone, and thus can control when unconventional protein secretion occurs’, can help readers understand the system at the abstract level.
Materials and Methods section:
- I understand that the design of constructs and strains have been demonstrated in previous studies of the authors. However, for readers who are not familiar with the system, it would be worth to briefly mention the rationale of genetic manipulation. For example, why did author need to manipulate mat B locus (maybe to control growth form), ip and upp1 loci (maybe they are common sites for genomic integration for transgenic production)?
- Please mention abbreviations used in the Table 2, both in main text and in Table caption. For example, CbxR: carboxin resistance, HygR: hygromycin resistance. I did not see PhleoR and NatR mentioned in the main text. It would be also useful to mention what Don3* is in the Materials and Methods section.
Section 2.4: I would prefer using ‘subculture’ or ‘transfer’ instead of ‘regenerate’. The word ‘regenerate’ may confuse readers as it has few other meanings.
Section 2.7: What is αGfbNB? Should it be aGfpNB?
Section 3.1:
- What do the authors mean by this? “strains grew yeast-like without any morphological phenotype both in glucose and in arabinose-containing media.” Probably ‘different morphological phenotype’?
- Do the authors have any explanation why signals are weaker in jps1 mutant when culturing in arabinose but not glucose? Did the author check the amount of produced proteins when culturing in glucose versus arabinose?
Section 3.2:
- “Gus reporter assays revealed that the lack of Don3 or Don3-Gfp fusion protein let to diminished extracellular Gus activity in glucose containing medium while a high activity occurred when the culture was grown in arabinose (Fig. 3C).” I think this sentence is confusing. Should “revealed that the lack of Don3 or Don3-Gfp fusion” be replaced with something like “revealed that the insertion of Don3 or Don3-Gfp fusion”?
- When the authors state “Culture supernatants of these arabinose cultures revealed the presence of free Gfp, suggesting that the full-length protein is secreted but largely degraded in the extracellular space. This is likely due to the presence of secreted proteases, a well-known phenomenon in fungi including U. maydis [15,25].”, I am curious that if protein degradation happens, why the signal of Gfp is still very strong, and stronger than cell extract fractions. Are the bands from Western blots analyses semi-quantitative between intracellular and extracellular fractions?
- When the authors say “In arabinose containing medium, cells grew normal and Gfp fluorescence localized to fragmentation zones indicating the fusion protein is only produced during ”on” conditions (Fig. 3B) [25]. Of note, in contrast to deregulated Jps1-Gfp, deregulated Don3-Gfp did not show any obvious mis-localization (Fig. 3B).”, I think it is worth to have another supplementary Figure that measures proportion of Don3 localization, like in Figure S1. This way the authors can clearly illustrate ‘no obvious mislocalization’.
Figure 3D: Maybe arrowheads that label protein bands are shifted? Please check.
Figure S4 legend: maybe the graph is for figs. 3C and 4C, not 3D and 4D?
Section 3.4:
- For “Cultures pre-grown in glucose showed a high level of induction two hours post medium switch (light blue columns),”, I would recommend saying “two hours after medium switch”.
- For “when cell harvest is conducted within the first hours after induction.”, I would recommend saying “the first few hours after induction”. Same with “…and thus showed only very weak extracellular Gus activities in the first hours after indu-tion (white columns).”
- I am quite skeptical with this statement, “By comparison, cultures grown overnight under constitutive induction in arabinose had no intracellular storage of Don3* and thus showed only very weak extracellular Gus activities in the first hours after induction (white columns).” and Figure 5. Have the authors checked the intracellular amount of Don3* to demonstrate that Don3* amount is much lower in the arabinose -NA-PP1 treatment compared to other treatments? If the authors say that there is no intracellular storage of Don3* in the arabinose -NA-PP1 treatment, the glucose treatment theoretically has no intracellular storage too. Why did the glucose treatment recover the signal faster than the arabinose -NA-PP1 treatment?
I envision this in another way. Only the arabinose -NA-PP1 treatment has a functional Don3*, and thus cell separation would happen the most. Meanwhile, the other two treatments would mostly have cell aggregates in cell population because there is no functional Don3* (either no protein or having protein but not functional). After the cultures were transferred to fresh arabinose media, the glucose and arabinose +NA-PP1 treatments can have functional Don3* to complete secondary septum formation right away, and the secretion of the gus-cts1 fusion protein is thus recovered. Meanwhile, as most cells in the arabinose -NA-PP1 treatment are completely separated, they require longer time to form new buds before creating septum and undergoing unconventional secretion. This results in more delayed signals in the arabinose -NA-PP1 treatment. Do the authors have data about cell population between treatments here (proportion of cell aggregates between three treatments)? This would certainly help explain what the authors see in Figure 5.
Section 3.5:
- When the authors state “In the presence of different ratios of mixed glucose and arabinose, cultures consumed the preferred carbon-source glucose first and switched to arabinose later, presumably, when the respective amount of glucose was completely metabolized.”, did the authors really check in the bioreactor that glucose is depleted first? I also think that using mols of sugars, instead of % w/v, may help the author figure out the consumption and growth as the molecular weights of glucose and arabinose are quite different (180.16 vs 150.13).
-Probably increasing sugar concentration does not yield higher biomass or fluorescence due to other limiting factors like depleted nitrogen sources, wastes in the system, etc.
Figure S7: Please check the figure title ‘Western blots of if different NB-Cts1 secreting candidates’.

Author Response
Answers to the reviewer 1’s comments
We thank all three reviewers for their support and indicate their comments in bold font and our answers in normal red font. Indicated line numbers refer to lines in “Track Changes” document to visualize all implemented major text changes.
This manuscript attempts to develop a regulatory system for unconventional protein secretion, using U. maydis as a model organism. The authors utilized the lock-type mechanism during yeast bud separation to control protein export through the regulation of the septation factor Don3, which is required for Cts1 secretion. The heterologous protein can then be fused with Cts1 for production and export to culture media, which reduce costs and complexity of further protein utilization.
Overall, I think this study is really fascinating. The authors utilize basic knowledge of Fungal Cell Biology to develop a novel method for producing secreted proteins without perturbation from conventional protein secretion. The study design is elaborated, and the writing (esp. Results and Discussion) is easy to understand the flow of the study. Thus, I think this manuscript is worth considering for publication.
I have several comments for authors that may improve the quality of their manuscript. As the manuscript file I received did not indicate line numbers, I would provide comments by sections.
We are sorry and now indicate line numbers.
Title: I think it would be useful to indicate that the authors work with U. maydis. This is to help audience have an idea about the scope of the paper. It also brings an attention that this unique system in U. maydis is utilizable while other eukaryotic systems are still not due to obscure knowledge.
We addressed this comment and a comment of reviewer 2 suggesting a complete change of the title by the following compromise: “Controlling unconventional secretion for production of heterologous proteins in Ustilago maydis through transcriptional regulation and chemical inhibition of kinase Don3”. We decided to keep the information of heterologous protein production in the title as it constitutes an important point in the manuscript.
Abstract: I would recommend mentioning Don3 as it is a key player for controlling protein export in this study. Brief explanation, like saying ‘Regulation of Don3 can delay the formation of fragmentation zone, and thus can control when unconventional protein secretion occurs’, can help readers understand the system at the abstract level.
We like this suggestion. We therefore added a sentence on the role of Don3 for Cts1 secretion and explained how it is exploited for regulation of unconventional secretion.
Materials and Methods section:
- I understand that the design of constructs and strains have been demonstrated in previous studies of the authors. However, for readers who are not familiar with the system, it would be worth to briefly mention the rationale of genetic manipulation. For example, why did author need to manipulate mat B locus (maybe to control growth form), ip and upp1 loci (maybe they are common sites for genomic integration for transgenic production)?
We added information on the strategy of genomic integration at the ip and upp1 loci (Line 153 ff). We also explained the genetic details of the background strain AB33 which has been manipulated at the b locus to control the growth form (Line 148 ff).
- Please mention abbreviations used in the Table 2, both in main text and in Table caption. For example, CbxR: carboxin resistance, HygR: hygromycin resistance. I did not see PhleoR and NatR mentioned in the main text. It would be also useful to mention what Don3* is in the Materials and Methods section.
We introduced all abbreviations of the table now in the caption and as footnotes (Line 183 ff).
Section 2.4: I would prefer using ‘subculture’ or ‘transfer’ instead of ‘regenerate’. The word ‘regenerate’ may confuse readers as it has few other meanings.
We adapted this.
Section 2.7: What is αGfbNB? Should it be aGfpNB?
We corrected that. Furthermore, to not confuse the reader we now call the nanoboby directed against Gfp “anti-Gfp” instead of α-Gfp.
Section 3.1:
- What do the authors mean by this? “strains grew yeast-like without any morphological phenotype both in glucose and in arabinose-containing media.” Probably ‘different morphological phenotype’?
Correct, we adapted this.
Do the authors have any explanation why signals are weaker in jps1 mutant when culturing in arabinose but not glucose? Did the author check the amount of produced proteins when culturing in glucose versus arabinose?
Unfortunately, we do not have a good explanation for that. Data from Hartmann et al., suggest, that the Poma promoter does not show differences in activity when strains are cultured in glucose or arabinose [1]. Thus, it is unlikely, that the difference is based on differential expression of the fusion gene controlled by this strong constitutive promoter. Western blot analyses have not been performed yet. We could try this using anti-Gfp antibodies, but this was not possible in the short time provided for revision. We will however follow up on that in the future, because we are planning to revisit Jps1 regulation with other regulatory systems, hoping that it will turn out to be functional with different induction strategies.
[1] Hartmann, H. A., Krüger, J., Lottspeich, F., & Kahmann, R. (1999). Environmental signals controlling sexual development of the corn smut fungus Ustilago maydis through the transcriptional regulator Prf1. The Plant Cell, 11(7), 1293-1305.
Section 3.2:
- “Gus reporter assays revealed that the lack of Don3 or Don3-Gfp fusion protein let to diminished extracellular Gus activity in glucose containing medium while a high activity occurred when the culture was grown in arabinose (Fig. 3C).” I think this sentence is confusing. Should “revealed that the lack of Don3 or Don3-Gfp fusion” be replaced with something like “revealed that the insertion of Don3 or Don3-Gfp fusion”?
We agree and changed the sentence:
“Reporter assays revealed induction levels of extracellular Gus activity ranging between five- and seven-fold for Don3-Gfp and Don3, respectively, indicating efficient transcriptional regulation (Fig. S3B).” (Line 398)
- When the authors state “Culture supernatants of these arabinose cultures revealed the presence of free Gfp, suggesting that the full-length protein is secreted but largely degraded in the extracellular space. This is likely due to the presence of secreted proteases, a well-known phenomenon in fungi including U. maydis [15,25].”, I am curious that if protein degradation happens, why the signal of Gfp is still very strong, and stronger than cell extract fractions. Are the bands from Western blots analyses semi-quantitative between intracellular and extracellular fractions?
Unfortunately, it is not possible to directly compare the samples of cell extracts and culture supernatant: while we applied 10 µg total isolated protein for cell extracts, culture supernatants are precipitated and the protein in one lane corresponds to 50 ml culture supernatant. However, strong signals for Gfp can be expected albeit high proteolytic activity in the supernatant, because the protein is very stable due to its beta-barrel structure and thus, it enriches in the culture broth [2]. This also refers to a similar question of reviewer 3 concerning the presence of proteases cleavage sites in the Don3-Gfp fusion protein.
We adapted the text and it now reads:
”Culture supernatants revealed the presence of free Gfp, suggesting that the full-length protein is secreted into the extracellular space where Don3 is quickly degraded (Fig. 3B, Fig. S3C, D). This is likely caused by secreted proteases, a well-known phenomenon in fungi including U. maydis [15,25]. The high stability of the remaining Gfp is presumably due to its robust beta-barrel structure (Chiang et al 2001).” (Line 407)
[2] Chiang, C. F., Okou, D. T., Griffin, T. B., Verret, C. R., & Williams, M. N. (2001). Green fluorescent protein rendered susceptible to proteolysis: positions for protease-sensitive insertions. Archives of biochemistry and biophysics, 394(2), 229-235.
- When the authors say “In arabinose containing medium, cells grew normal and Gfp fluorescence localized to fragmentation zones indicating the fusion protein is only produced during ”on” conditions (Fig. 3B) [25]. Of note, in contrast to deregulated Jps1-Gfp, deregulated Don3-Gfp did not show any obvious mis-localization (Fig. 3B).”, I think it is worth to have another supplementary Figure that measures proportion of Don3 localization, like in Figure S1. This way the authors can clearly illustrate ‘no obvious mislocalization’.
The reviewer addresses an interesting point. Quantification of Jps1-Gfp localization was conducted after the authors detected an abnormal, so far not observed localization of Jps1. In previous studies, a localization of Jps1 restricted to the fragmentation zone was observed [3]. The regulated version now also resided in cytoplasmic aggregates. This prompted us to do the quantification.
For Don3 the situation is different: Its localization in the fragmentation zone was described before in a study of the septation mechanism [4 , (see Figure R1 top panel)]. We observed the exact same localization pattern in both of our Don-Gfp constructs (wild type and ATP-analogue sensitive version) without any further unexpected patterns. More specifically, we never came across a divergent localization in several microscopical analyses of the strains (examples in Figure R1). Therefore, we believe it is not essential to conduct a detailed quantification for this example. In addition, this would require the generation of an appropriate control strain with the native regulation. This would take about 6 weeks in U. maydis and is not achievable in the time provided for revision. We however tried to be more precise in our wording to solve this issue and the text now reads:
“Deregulated Don3-Gfp solely localized to fragmentation zones of dividing cells (Fig. 3B, S3A) which is identical to published results (Böhmer et al., 2008).” (Line 396)
Figure R1: Top panel, published localization of Don3-Gfp in the fragmentation zone of dividing cells [4]. Lower panels, different microscopic picture taken during the work on the manuscript showing that no unexpected localization outside the fragmentation zone occurred for neither of both versions of Don3-Gfp.
[3] Reindl, M., Stock, J., Hussnaetter, K. P., Genc, A., Brachmann, A., & Schipper, K. (2020). A novel factor essential for unconventional secretion of chitinase Cts1. Frontiers in microbiology, 11, 1529.
[4] Böhmer, C., Böhmer, M., Bölker, M., & Sandrock, B. (2008). Cdc42 and the Ste20-like kinase Don3 act independently in triggering cytokinesis in Ustilago maydis. Journal of cell science, 121(2), 143-148.
Figure 3D: Maybe arrowheads that label protein bands are shifted? Please check.
The reviewer is right, we corrected that.
Figure S4 legend: maybe the graph is for figs. 3C and 4C, not 3D and 4D?
Thank you! This is true and we corrected it.
Section 3.4:
- For “Cultures pre-grown in glucose showed a high level of induction two hours post medium switch (light blue columns),”, I would recommend saying “two hours after medium switch”.
Corrected.
- For “when cell harvest is conducted within the first hours after induction.”, I would recommend saying “the first few hours after induction”. Same with “…and thus showed only very weak extracellular Gus activities in the first hours after indu-tion (white columns).”
Corrected.
- I am quite skeptical with this statement, “By comparison, cultures grown overnight under constitutive induction in arabinose had no intracellular storage of Don3* and thus showed only very weak extracellular Gus activities in the first hours after induction (white columns).” and Figure 5. Have the authors checked the intracellular amount of Don3* to demonstrate that Don3* amount is much lower in the arabinose -NA-PP1 treatment compared to other treatments? If the authors say that there is no intracellular storage of Don3* in the arabinose -NA-PP1 treatment, the glucose treatment theoretically has no intracellular storage too. Why did the glucose treatment recover the signal faster than the arabinose -NA-PP1 treatment?
I envision this in another way. Only the arabinose -NA-PP1 treatment has a functional Don3*, and thus cell separation would happen the most. Meanwhile, the other two treatments would mostly have cell aggregates in cell population because there is no functional Don3* (either no protein or having protein but not functional). After the cultures were transferred to fresh arabinose media, the glucose and arabinose +NA-PP1 treatments can have functional Don3* to complete secondary septum formation right away, and the secretion of the gus-cts1 fusion protein is thus recovered. Meanwhile, as most cells in the arabinose -NA-PP1 treatment are completely separated, they require longer time to form new buds before creating septum and undergoing unconventional secretion. This results in more delayed signals in the arabinose -NA-PP1 treatment. Do the authors have data about cell population between treatments here (proportion of cell aggregates between three treatments)? This would certainly help explain what the authors see in Figure 5.
In our earlier publication, we had analysed Don3 in more detail and found that it is unconventionally secreted, like Cts1, during the cell cycle [5]. Since it is known that it localizes to the fragmentation zone, we concluded that it is released in a similar manner as observed for Cts1. Based on this observation, we think that the intracellular pool of Don3 is depleted regularly (i.e. upon every cell separation event). Hence, when cell separation is abolished upon NA-PP1 treatment or in glucose, we expect the protein to accumulate while the active protein promotes septum formation and its own depletion during cell separation in cultures grown in arabinose without inhibitor. We did not mention that Don3 is secreted in the text, which might have been confusing to the readership. We now added this information and hope that the issue is clearer now.
In addition, we also think that the explanation of the reviewer is correct with the effect of the cell morphology and adds on this effect. Both in glucose and arabinose + NA-PP1 cultures cells are present in aggregates. Upon induction there are thus many cells that start budding off at the same time as compared to the arabinose – NA-PP1 culture. We also added this hypothesis in the text. The section now reads:
“By comparison, cultures grown overnight under constitutive induction in arabinose had no intracellular storage of Don3* due to its unconventional secretion during cell separation (Aschenbroich 2019). These cultures and thus showed only very weak extracellular Gus activities in the first few hours after induction (white columns). They reached a comparable level to the other cultures only after four hours. After 8 hours, all cultures exhibited extracellular Gus activities, which were not significantly different from each other anymore. The difference in immediate induction levels between the preincubated culture in arabinose lacking NA-PP1 and those preincubated in glucose or arabinose with NA-PP1 might be further boosted by the fact that the latter cells are present in aggregates prior to induction and all these start budding at the same time after induction.” (Line 552 ff)
We have quantified cell aggregates earlier; however, conducting this experiment in triplicates requires a substantial amount of time, which we did not have for the revision. The same is true for a quantification via Western blot.
[5] Aschenbroich, J., Hussnaetter, K. P., Stoffels, P., Langner, T., Zander, S., Sandrock, B., Bölker, M., Feldbrügge, M. & Schipper, K. (2019). The germinal centre kinase Don3 is crucial for unconventional secretion of chitinase Cts1 in Ustilago maydis. Biochimica et Biophysica Acta (BBA)-Proteins and Proteomics, 1867(12), 140154.
Section 3.5:
- When the authors state “In the presence of different ratios of mixed glucose and arabinose, cultures consumed the preferred carbon-source glucose first and switched to arabinose later, presumably, when the respective amount of glucose was completely metabolized.”, did the authors really check in the bioreactor that glucose is depleted first?
We indirectly checked on that because gfp expression (and hence, fluorescence of the Gfp protein) is abolished in the presence of glucose when the Pcrg promoter is used [6]. We also know that glucose is the preferred carbon source as this has been shown in a PhD thesis from a collaborator at RWTH Aachen [7].
[6] Bottin, A., Kämper, J., & Kahmann, R. (1996). Isolation of a carbon source-regulated gene from Ustilago maydis. Molecular and General Genetics MGG, 253(3), 342-352.
[7] Online Analytics of Pectic Compound Degradation in Small-Scale Using Ustilago maydis, Markus Müller, 2019, http://publications.rwth-aachen.de/record/774743/files/774743.pdf).
I also think that using mols of sugars, instead of % w/v, may help the author figure out the consumption and growth as the molecular weights of glucose and arabinose are quite different (180.16 vs 150.13).
This is a good point. As suggested by the reviewer, we now provide the concentrations (molarities) in addition to the g/L values.
-Probably increasing sugar concentration does not yield higher biomass or fluorescence due to other limiting factors like depleted nitrogen sources, wastes in the system, etc.
We agree that this a possibility and added this hypothesis in the text. (Line 655)
Figure S7: Please check the figure title ‘Western blots of if different NB-Cts1 secreting candidates’.
We corrected that.
Reviewer 2 Report
This paper describes a system of contolling heterolgous protein excreting using the unconventional system deployed in Ustilago maydis by manipulating the transcriptional and translational expression of the proteins involved.
The methods are scientifically sound and the experiments are appropriately duplicated and statistically verified.
Overall, the manuscript is descriptive and walks the reader through the deductive process of understanding how to best control protein secretion in this system. However, the paper is too verbose and overly repeats and cites studies conducted by the authors in previous publications. The main results of the study are lost in the details, including the post-translational inhibition of Don3 by ATP analogs for culture control. The main results are also not highlighted in the abstract or outlined in the introduction. In essence, the paper lacks focus and needs to be streamlined significantly perhaps under a new title of "Controlling the unconventional secretion system in U. maydis in culture through chemical inhibition of Don3 and transcriptional regulation".
I am recommending "accept after minor revision" even though my comments suggest a major overhaul of the manuscript. My reasoning for this is:
- The experiments are conducted appropriately and no controls are missing.
- Although frustrating, it is much easier to streamline and make a manuscript more concise when all of the research is done.
- The authors can easily move large sections of the manuscript to the supplementary information and only highlight the main findings of the manuscript.
- Experiments that merely support previous findings should be eliminated from the manuscript.
- 16/63 citations are self-citations. I suggest the authors cite only the most recent publications they have on this unconventional secretion system.
- The results and the discussion should be written separately. The main results should highlight new findings while the discussion allows for the elaboration of how this system might be more cost effective due to no media changes (unless for pharmaceutical grade products).
Author Response
Dear reviewer 2,
We thank you for your thorough reading of our manuscript. Please find attached the author’s reply to the review report.

Reviewer 3 Report
In this manuscript submitted by Hussnaetter et al., the authors attempted to establish the condition of heterologous protein secretion via unconventional secretory pathway in the model fungus Ustilago maydis. Using the promoters responsive to nutrient and protein kinase variant, they established the regulation system by transcriptional and post-transcriptional manners. Eventually, the authors nicely succeeded to produce nanobodies.
Overall, I could have an impression that the manuscript is already in good shape for publication, although the text seems a little bit tedious. I found a few minor errors, so I would like to ask the authors to correct them.
In Materials and Methods, the authors use "hydrolyze" to cut plasmid by restriction enzymes. The word "hydrolyze" is not so common. "digest" would be more better.
Page 6, "2% (w/v) agar agar." <- two times "agar". One should be deleted.
Figure 3D, right side labels seems to be out of alignment.
From Figure 3D and Figure 4D, Don3-GFP or its variant protein seems to be cleaved and only GFP is secreted. Are there any cleavage site at the C-terminus of Don3 like Kex2 ?
Author Response
Dear reviewer 3,
We thank you for your thorough reading of our manuscript. Please find attached the author’s reply to the review report.

Round 2
Reviewer 1 Report
Comments for the authors
Thank you very much for considering my suggestions and providing thorough responses. I completely understand that parts of data require significant amount of time to produce, and that could not be achieved during the revision process. Some of those questions are just from my curiosity. However, the authors have done a wonderful job to address the comments, with supports from previous studies. I think this manuscript is worth for publication to the Journal of Fungi.
Just a couple of final minor comments for the authors:
- For the title, I recommend saying “Don3 kinase” or “the kinase Don3”.
- Abstract Line 15: I recommend saying “Don3 is an essential kinase for functional assembly …” or “The kinase Don3 is essential for functional assembly….”.
Author Response
Dear Reviewer 1,
thank you very much for your positive feedback! We adapted your two recommendations and uploaded the final version as .pdf file and all revised versions of the manuscript in one .zip folder.
Best regards,
The authors